# Are Posterior Oropharyngeal Saliva Specimens an Acceptable Alternative to Nasopharyngeal Sampling for the Monitoring of SARS-CoV-2 in Primary-Care Settings?

**DOI:** 10.3390/v13050761

**Published:** 2021-04-26

**Authors:** Shirley Masse, Camille Bonnet, Ana-Maria Vilcu, Hayat Benamar, Morgane Swital, van der Werf Sylvie, Fabrice Carrat, Thomas Hanslik, Thierry Blanchon, Alessandra Falchi

**Affiliations:** 1UR7310 Bioscope, Université de Corse Pascal Paoli, 20250 Corte, France; masse_s@univ-corse.fr; 2Sorbonne Université, INSERM, Institut Pierre Louis d’Epidémiologie et de Santé Publique, (IPLESP), F-75012 Paris, France; camille.bonnet@iplesp.upmc.fr (C.B.); ana-maria.vilcu@iplesp.upmc.fr (A.-M.V.); hayat.benamar@iplesp.upmc.fr (H.B.); morgane.swital@iplesp.upmc.fr (M.S.); fabrice.carrat@iplesp.upmc.fr (F.C.); thomas.hanslik@aphp.fr (T.H.); thierry.blanchon@iplesp.upmc.fr (T.B.); 3Unité de Génétique Moléculaire des Virus à ARN, Institut Pasteur, 75015 Paris, France; svdwerf@pasteur.fr; 4Centre Coordonnateur du Centre National de Référence des Virus des Infections Respiratoires (Dont la Grippe), Institut Pasteur, 75015 Paris, France; 5Unité de Génétique Moléculaire des Virus à ARN, UMR CNRS 3569, 75015 Paris, France; 6Sorbonne Paris Cité, Unité de Génétique Moléculaire des Virus à ARN, Université Paris Diderot, 75015 Paris, France; 7AP-HP, Hôpital Saint-Antoine, Unité de Santé Publique, 75012 Paris, France; 8UFR de Médecine, Université de Versailles Saint-Quentin-en-Yvelines, UVSQ, 78000 Versailles, France; 9Assistance Publique—Hôpitaux de Paris APHP, Hôpital Ambroise Paré, Service de Médecine Interne, 92100 Boulogne Billancourt, France

**Keywords:** saliva, SARS-CoV-2, respiratory tract infections, primary health care, sentinel surveillance

## Abstract

Background: The present study was set up to evaluate the efficacy of virological surveillance using posterior oropharyngeal saliva samples to monitor the COVID-19 pandemic in general practice. Methods: Posterior oropharyngeal saliva samples were collected without restriction on timing or alimentation by general practitioners from patients with acute respiratory infection (ARI) seen in consultation. Saliva samples were tested by real-time reverse transcription polymerase chain reaction for SARS-CoV-2 and 21 other respiratory pathogens. Results for SARS-CoV-2 in saliva samples were compared to results obtained using a nasopharyngeal swab (NPS) collected in a certified medical laboratory before or after the ARI consultation. Results: Overall, 143 ARI patients were enrolled between 6 June 2020, and 19 January 2021. SARS-CoV-2 RNA was detected in 37.0% (n = 53) of saliva samples and in 39.0% (n = 56) of NPS. Both saliva and NPS were positive in 51 patients. Positive and negative results were concordant between saliva samples and NPS in 51 (96.2%) and in 85 (94.4%) patients, respectively, with a Cohen’s Kappa coefficient of 0.89 (95% CI 0.82–0.97, *p* < 0.001). Other respiratory viruses were detected in 28.0% (n = 40) of the 143 saliva samples. Conclusions: Findings suggest that saliva samples could represent an attractive alternative to NPS for surveillance of SARS-CoV-2 in patients consulting for an ARI in primary care.

## 1. Introduction

Reverse transcription–quantitative polymerase chain reaction (RT–qPCR) of nasopharyngeal samples is widely used to detect SARS-CoV-2 and other respiratory viruses [1,2]. However, collecting nasopharyngeal samples (NPS) causes patient discomfort, which may discourage patients with mild symptoms from seeking diagnostic tests. This could compromise the detection of cases and epidemiological surveillance.

The French Sentinelles network monitors acute respiratory infection (ARI) in primary health care in France from epidemiological and virological data collected by general practitioners (GPs) and pediatricians. This allows to detect and characterize the circulating respiratory viruses, as SARS-CoV-2, influenza and RSV [3]. From the beginning of the COVID-19 pandemic, Sentinelles physicians encountered difficulties conducting virological ARI surveillance for patients with suspected COVID-19 cases using NPS, due to the following two recommendations in France: systematic SARS-CoV-2 virological test (NPS) should be conducted in specific diagnosis centers and the requirement to receive the result very quickly. The consequence is the reluctance for Sentinelles physicians and the patients to perform a NPS for ARI surveillance in addition to the NPS done in a COVID-19 diagnosis center. These issues have led to a decrease in the number of patients included in the ARI virological surveillance since the start of the COVID-19 pandemic. Therefore, it seems crucial to propose an alternative method to NPS to improve the virological monitoring of ARIs in primary health care.

Saliva samples for COVID-19 diagnosis have been evaluated in previous studies [4,5,6,7]. To better understand the efficacy of these type of samples in a field situation, such as the monitoring of circulating respiratory viruses, we conducted a prospective study to estimate the performance of saliva samples self-collected by the patient at the Sentinelles physicians’ offices. The present study was set up to compare the relative sensitivity of NPS and saliva samples used in a surveillance strategy for detection of SARS-CoV-2 for the initial diagnosis of mild cases seen in primary care surveillance settings. The results would allow progress in the management of SARS-CoV-2 infection and other respiratory viruses through easy and rapid collection of saliva specimens in the field, which would facilitate and improve epidemiological surveillance in primary care. We also used saliva specimens to identify respiratory viruses associated or not associated with SARS-CoV-2 infection.

## 2. Materials and Methods

### 2.1. Study Design

This study was conducted by GPs and pediatricians of the French Sentinelles Network [8]. The patients aged eight years or older consulting for an ARI episode, who had already undergone a NPS in a clinical laboratory or who intended to undergo it after the consultation, were invited to participate. ARI was defined as “sudden onset of fever (or feeling feverish), accompanied by respiratory signs”.

Sociodemographic data, clinical characteristics and date of symptom onset were collected by Sentinelles physicians for each patient at inclusion. The patients were asked to provide a self-collected saliva sample while at the medical office. In particular, participants were asked to produce saliva coughed up from the posterior oropharynx by clearing the throat and/or by gargling for 15–20 s with 1 mL of water and spitting it back into a tube containing 1 mL of viral transport medium. Saliva was collected without restriction on timing or intake of food to simplify the collection protocol. A visual guide was provided and physician training was conducted. Saliva samples were triple-sealed and transported by mail to the virology laboratory within 24 h.

A NPS was performed in a clinical laboratory before or after the ARI consultation, according to the national testing and contact-tracing strategy. Clinical laboratories transmitted RT–qPCR results to physicians and patients as positive, negative, or uncertain. The Ct values of positive NPS were not recorded. Sentinelles physicians transmitted the date of NPS collection and the RT–qPCR result for each included patient to the study team.

### 2.2. Outcome

We included in the analysis only patients having undertaken both the saliva and the NP samples within 9 days of symptoms onset, because (i) the RT–qPCR sensitivity of NPS is >70% within this time range, with a peak of >90% during the first 5 days, (ii) previous studies indicated that the sensitivity in saliva reached 90% in individuals with symptoms ≤9 days duration [9,10], and (iii) infectiousness duration and subsequent virus isolation timelines could be counted for 9 days from symptom onset in mild cases [11,12]. To ensure the presence of SARS-CoV-2 RNA and to avoid false positives, a sample was defined as positive when positive test results were obtained for more than one genetic locus and assay, and all others were defined as negative [13].

### 2.3. SARS-CoV-2 RNA Detection in Saliva Samples

Total nucleic acid was extracted from 200 μL of saliva samples using the QIACUBE processing system with the QIAamp 96 Virus QIAcube HT Kit (Qiagen, Hilden, Germany) and eluted into 100 μL of total nucleic acid. RT–qPCR was performed for each sample in the first instance with a TaqPath COVID-19 Kit (Thermo Fisher Scientific, Waltham, MA, USA). The TaqPath COVID-19 Kit is a multiplex RT–qPCR diagnostic assay targeting three regions of the SARS-CoV-2 genome (N, S, and ORF1ab), which was approved by the French National Center of Respiratory Diseases for the detection of SARS-CoV-2. To ascertain the accuracy of the TaqPath COVID-19 kit results, results were confirmed retrospectively using the in-house method reported by Pezzi et al. [14]. Using a dual-target assay can help prevent false-negative results because of polymorphisms, point mutations, or major sequence deletions/insertions. In this in-house method, two single-target assays recommended by the World Health Organization (E-sarbeco envelope gene (Charite University, Berlin, Germany) and RdRp-IP4 (RdRp, Institut Pasteur, Paris, France)) were selected and combined in a uniquely robust test allowing the detection of these two regions in a single FAM channel [14]. These two assays (TaqPath COVID-19 and the in-house method) exhibited a specificity of 100% [13]. An MS2 internal control for nucleic acid extraction was available for both assays.

Saliva samples were also analyzed using the Bosphore Respiratory Pathogens Panel Kit v6. This CE-certified 7-tube multiplex quantitative PCR kit allowed us to detect the following 21 respiratory pathogens: influenza A; influenza B; A(H1N1)pdm09; respiratory syncytial virus (RSV) A/B; parainfluenza (PIV) 1, 2, 3, and 4; enterovirus; metapneumovirus; bocavirus; rhinovirus (HRV); coronavirus (CoV) OC43, NL63, HKU1, and 229E; adenovirus; *Haemophilus influenzae* type B; parechovirus; and *Mycoplasma pneumoniae*.

### 2.4. Statistical Analysis

Descriptive statistics were presented as number (%) for categorical variables and as means ± standard deviation or median (interquartile range; IQR) for continuous variables. The Cohen’s kappa coefficient was used to estimate the agreement between NPS and saliva SARS-CoV-2 results. Kappa values denote the following levels of agreement: Poor agreement = < 0.20, fair agreement = 0.20–0.40, moderate agreement = 0.40–0.60, good agreement = 0.60–0.80, and very good agreement = 0.80–1.00 [15]. We also assessed the association between time from symptom onset and Ct of saliva samples by using a simple linear regression. The R function lm() was used to compute linear regression model [16,17]. A *p*-value < 0.05 was considered significant. All analyses were performed using R, version 4.0.0 (R statistical computing).

### 2.5. Ethics Approval

The protocol was approved by the French Data Protection Agency (CNIL#920211) and the French ethics research committee (CPP#47/20). Consent was obtained from swabbed patients.

## 3. Results

### 3.1. Patient Characteristics

Overall, 184 patients were enrolled by Sentinelles physicians between 6 June 2020, and 19 January 2021, and 143 were included in the final analysis (Figure 1). The characteristics of these patients are described in Table 1. The majority were women (n = 80, 56.0%). The median age of the patients was 35 years (range: 8–74 years, IQR: 22.5–49), 21.0% (n = 30) had at least one chronic disease. The median delay between onset of symptoms and sample collection (saliva or NPS) was two days (IQR 1–3 for saliva samples and 1.5–3 for NPS samples). The median delay between NPS and saliva sample collection was 0 days (range –5 to 3 days) with 39.0% (n = 56) of samples collected the same day (on day of inclusion). Most patients were included between September and November 2020, during the second COVID-19 wave.

### 3.2. Virological Results

Of the 143 patients, SARS-CoV-2 RNA was detected in 39.0% (*n* = 56) of NPS and in 37.0% (*n* = 53) of saliva samples (Table 2). Both saliva and NPS were positive in 51 patients (2 patients were positive only in saliva samples and 5 only in NPS). Positive results were concordant between saliva samples and NPS in 51 patients (96.2%) and negative results were concordant between saliva samples and NPS in 85 (94.4%) with a Cohen’s Kappa coefficient of 0.89 (95% CI 0.82–0.97, *p* < 0.001) (Table 3). This concordance trend did not vary significantly according to the number of days between the collection of NPS and saliva samples (data not shown) (*p* = 0.4).

All 143 samples (concordant and discordant) were re-extracted and retested using the in-house method describe above in the Materials and Methods. As reported in Table 3, in two patients SARS-CoV-2 RNA was detected in saliva samples only. These two saliva samples showed a Ct of 12 and 22 respectively. Among the five patients showing a positivity in SARS-CoV-2 RNA in NP samples only, three paired-saliva samples were negative for both RT–qPCR diagnostic assays used in this study. The other two paired-saliva samples showed a weak signal (Ct > 38) with the in-house method. Because a saliva sample was defined as positive when positive test results were obtained for more than one genetic locus and assay, these three samples were considered negative. The median Ct for detection of SARS-CoV-2 in saliva samples was 24 (IQR: 20–33). The Ct values for SARS-CoV-2 increased over time with respect to days since symptom onset, but with no significant difference (*p* = 0.78) (Figure 2).

Respiratory viruses other than SARS-CoV-2 were detected in 28.0% (*n* = 40) of the 143 saliva samples (Table 2). Viruses detected were HRV (25.0%; *n* = 36), HCoV HKU1 (2.8%; *n* = 4), and PIV1 (0.7%; *n* = 1). The median Ct for the detection of HRV, the main respiratory virus detected other than SARS-CoV-2, was 26 (IQR: 24–30). The Ct values for HRV increased significantly over time with respect to days since symptom onset (*p* value = 0.03) (Figure 2). Of the 53 patients who tested positive for SARS-CoV-2, three showed concurrent infection with at least one non-SARS-CoV-2 respiratory viral pathogen. Of these three patients, two were health-care assistants (SARS-CoV-2/PIV 1 and SARS-CoV-2/HCoV HKU1) (Table 2).

The positivity rate for all respiratory viruses detected in saliva samples ranged from 17.0% at day 1 since symptom onset to 43.0% at day 8 and peaked at day 5 (89.0%). This trend was similar for SARS-CoV-2 and HRV (Figure 3).

## 4. Discussion

The main findings of the present study suggested that saliva samples collected without restriction on timing or intake of food could represent an attractive alternative to NPS for surveillance of SARS-CoV-2 in symptomatic patients consulting for an ARI in primary care. The results also showed that almost one-third of saliva samples were positive for at least one respiratory pathogen other than SARS-CoV-2, strengthening the potential usefulness of saliva as a trustworthy virological surveillance tool for ARIs.

Since the beginning of the COVID-19 pandemic, several studies have estimated the sensitivity of saliva for detection of SARS-CoV-2 as moderate (69%) to high (100%) relative to NPS. The variation in sensitivity seen in these studies, mostly performed in hospitals, probably reflects differences in the severity of symptoms, timing of testing relative to symptom onset, method of saliva collection (cough saliva, posterior oropharyngeal saliva, saliva swab, or unstimulated saliva), avoidance or no avoidance of eating, drinking, or brushing teeth before collection, morning collection versus other times of the day, and the laboratory techniques used to analyze samples [7]. The results of the present study, showing a high agreement between saliva and NPS samples with respect to RNA SARS-CoV-2 detection, were similar to those previously obtained in symptomatic ambulatory patients presenting with mild symptoms [18], despite several specific constraints related to the collection of saliva samples in the field. Even if early-morning specimens tend to have a higher viral load than saliva samples collected at other times of the day, we collected saliva throughout the day because this was more consistent with the ARI surveillance protocol in the field [19]. To obtain an enhanced saliva sample [19,20], patients with ARI were instructed and supervised by physicians to produce saliva coughed up from the posterior oropharynx (by clearing the throat or gargle) as this kind of sample seems to be much more sensitive than saliva from the oral cavity alone or from salivary glands [21]. However, recent studies suggested that oral cavity saliva specimens and deep throat sputum offer high sensitivity and specificity too [22,23].

The failure to collect concomitant paired samples of saliva and NPS may have affected the concordance of SARS-CoV-2 results. Despite this, the concordance between the results for these two kinds of samples did not vary significantly according to the time lapse between collection and symptom onset. We sought to limit this important methodological bias by including only individuals with symptom onset ≤9 days, because the sensitivity of RT–qPCR is >70% within this time range for NPS and because in mild cases shedding of replication-competent virus appears to be rare ≥10 days after onset of symptoms [11,12,24]. Moreover, previous studies showed that the sensitivity of saliva testing increased to 90% in individuals with symptoms of <9 days duration [9,10].

In our study, the positivity rate of salivary samples seems to peak on the fourth and fifth day after the onset of symptoms for both SARS-CoV-2 and HRV. This is useful information to improve the timing of ARI monitoring protocol. To our knowledge, no study described the time course of the positivity rate of respiratory viruses other than SARS-CoV-2 in relation to symptom onset in saliva samples.

Saliva sampling has been suggested as a valuable tool for detection of respiratory viruses such as influenza A virus, influenza B virus, PIV, and RSV [25,26]. In the present study, we observed that nearly one-third of saliva samples were positive for viruses other than SARS-CoV-2. These findings are in agreement with data reported in southern France for NPS samples of patients enrolled from March to April 2020 that were analyzed for SARS-CoV-2 (15.2%) and non-SARS-CoV-2 respiratory viruses (25.9%) [27]. Our results suggest that coinfection with other respiratory pathogens could be detected in saliva samples. In the present study, coinfection was uncommon among patients with COVID-19; although the rate was consistent with that reported in previous studies reporting SARS-CoV-2 coinfection rates in NPS samples [28], it was lower than the rate of viral coinfection reported among patients with SARS-CoV-2 using respiratory virome characterization [29]. Determining the coinfection rate and its consequent clinical impacts on COVID-19 is critical, particularly where therapeutic interventions are available for some coinfecting agents such as influenza virus.

There are several limitations to our study. First, although the procedure of saliva sample collection was supervised by physicians, there was interindividual variability in the quality of the saliva sample. Second, the interpretation of NPS results is limited by the heterogeneity in the quality of samples, the sensitivity of the RT–qPCR assay used by clinical laboratories, and the absence of information about the Ct. Third, false positive/negative results may have occurred in NPS and saliva samples. However, testing for at least two targets and the low cycle threshold values observed make it likely that all saliva samples were true positives. We have not been able to ascertain this for NPS. Fourth, as we were not able to compare the detection rates of respiratory viruses other than SARS-CoV-2 in saliva and NPS, we could make no conclusion about the use of saliva samples for their detection. Finally, we collected saliva coughed up from the posterior oropharynx (by clearing the throat or gargle). However, recent studies suggested that oral cavity saliva specimens and deep throat sputum also offer high sensitivity and specificity [22,23].

Despite the limitations of the data collected, the main findings of this study showed that saliva samples could be an acceptable alternative to NPS for the virological surveillance of SARS-CoV-2 in mildly symptomatic patients with ARI in primary care. Systems for data collection on patients with ARI have been affected at a European level by the COVID-19 pandemic, leading to some difficulties in providing robust 2019–2020 estimations of influenza vaccination in primary care [30]. These findings suggest that the circulation of respiratory pathogens other than SARS-CoV-2 could be monitored by using saliva samples. In conclusion, a simple and noninvasive self-collection procedure such as saliva sampling could improve the data collection in primary care by avoiding the need for a swab, reducing exposure of health-care workers, and allowing self-collection at any time of the day. This has the potential to be used in outbreak settings where simultaneous testing of large numbers of people of all ages is warranted.

## Figures and Tables

**Figure 1 viruses-13-00761-f001:**
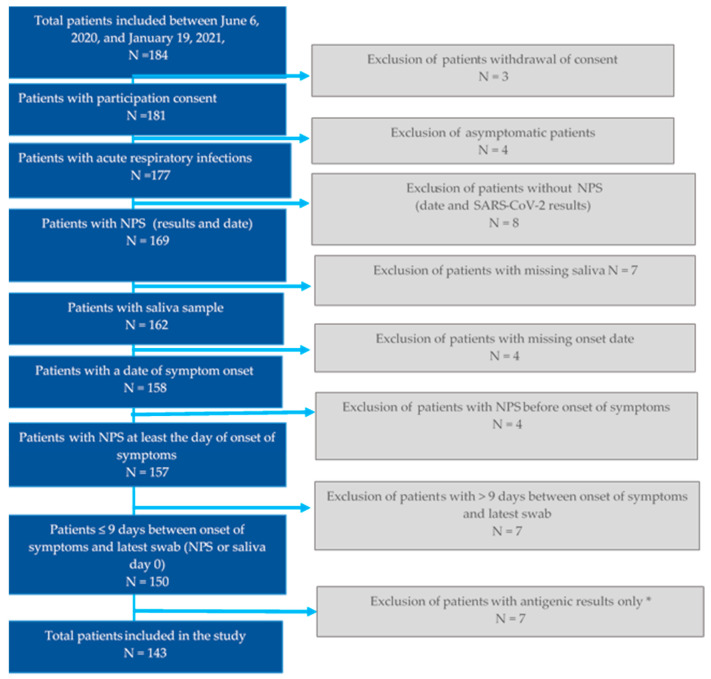
Flowchart of analysis of patients with ARI. ***** Patients tested with antigenic assay alone, were excluded as we compared paired-samples (NP and saliva) tested at a minimum by RT–qPCR.

**Figure 2 viruses-13-00761-f002:**
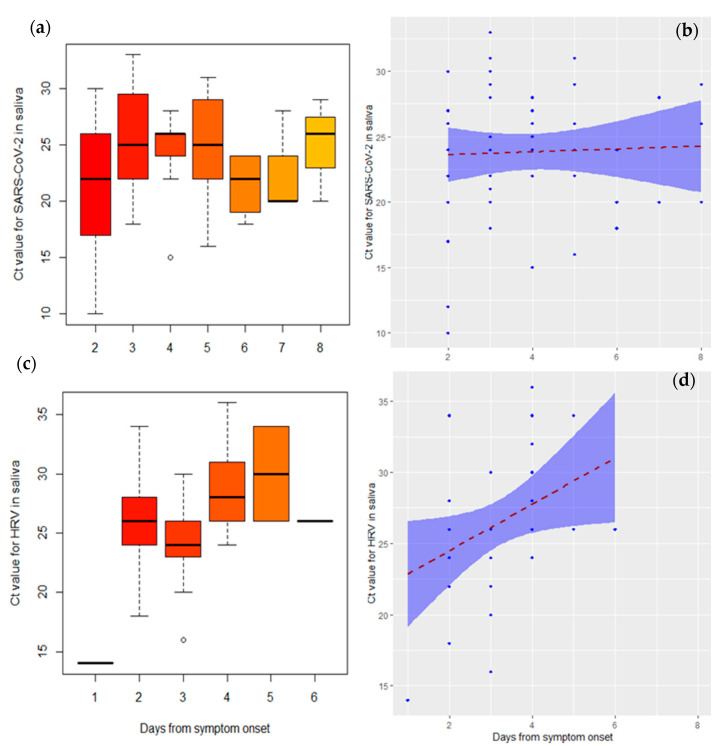
Boxplot and scatter plot of Ct values of SARS-CoV-2 (**a**,**b**) and HRV(**c**,**d**) saliva-positive patients with ARI. The R function lm() was used to compute linear regression model. The blue-shaded areas correspond to the 95% confidence limits. The number of samples positive to the respiratory viruses other than SARS-CoV-2 and HRV was too small to be plotted.

**Figure 3 viruses-13-00761-f003:**
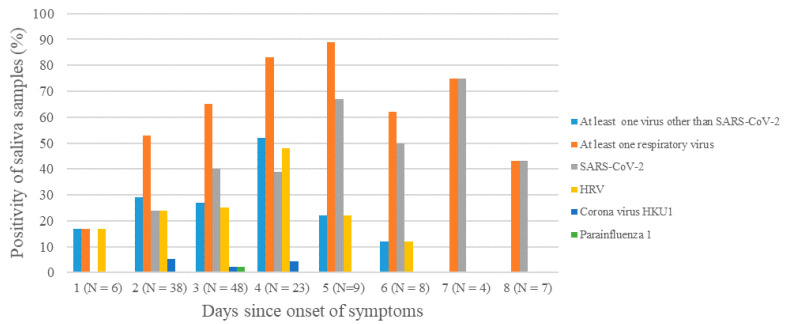
Positivity rates of saliva samples of ARI patients for SARS-CoV-2 and the respiratory viruses other than SARS-CoV-2 analyzed in the present study.

**Table 1 viruses-13-00761-t001:** Characteristics of patients with ARI included by the Sentinelles physicians.

Characteristics	*N* = 143
Sex (n; %)	
Female	80 (56.0%)
Age (mean), years	35.8 (8–74)
Age (median), years	35 (22.5–49)
Age group (years)	
[8,15)	20 (14.0%)
[15,35)	51 (36.0%)
[35,45)	26 (18.0%)
[45,55)	24 (17.0%)
[55,65)	15 (10.0%)
≥65	7 (4.9%)
At least one chronic condition (*n*; %)	
Yes	30 (21.0%)
Missing	1 (0.7%)
Obesity (*n*; %)	
Yes	21 (15.0%)
Missing	3 (2.1%)
Smoking (*n*; %)	
Yes	30 (21.0%)
Missing	3 (2.1%)
Pregnancy (*n*; %)	
Yes	1 (0.7%)
Missing	5 (3.3%)
Health professional (*n*; %)	
Yes	16 (11.0%)
Missing	3 (2.1%)
Collection month (*n*; %)	
June	8 (5.6%)
July	4 (2.8%)
August	1 (0.7%)
September	25 (17.5%)
October	46 (32.2%)
November	44 (30.8%)
December	8 (5.6%)
January	7 (4.9%)
Days between onset of symptoms and saliva sampling (*n*; %)	
1	6 (4.2%)
2	38 (27.0%)
3–4	71 (49.6%)
5–6	17 (11.8%)
7–8	11(7.7%)
Mean	2.4 (0–7)
Median	2 (1–3)
Days between onset of symptoms and NPS (*n*; %)	
1	11 (7.7%)
2	25 (17.0%)
3–4	77 (53.8%)
5–6	20 (13.9%)
7–9	10 (7.0%)
Mean	2.5 (0–8)
Median	2 (1.5–3)
Days between collection of NPS and saliva sampling (*n*; %)	
From –5 to –1	31 (21.7%)
0	56 (39.0%)
1	44 (31.0%)
2	11 (7.7%)
3	1 (0.7%)
Median	0 (0–3)

**Table 2 viruses-13-00761-t002:** Virological results for 143 samples (saliva and NPS) analyzed for SARS-CoV-2 and other respiratory pathogens.

Virological Results	*N* = 143
**Detection in NPS**	
SARS-CoV-2	56 (39.0%)
**Detection in saliva sample**	
Positive for at least one respiratory virus	90 (63.0%)
SARS-CoV-2	53 (37.0%)
Positive for at least one respiratory virus other than SARS-CoV-2	40 (28.0%)
HRV	36 (25.0%)
Human coronavirus HKU1	4 (2.8%)
Human parainfluenza 1	1 (0.7%)
Coinfection	4 (2.8%)
HRV, human coronavirus HKU1	1 (0.7%)
SARS-CoV-2, human coronavirus HKU1	1 (0.7%)
SARS-CoV-2, human parainfluenza type 1	1 (0.7%)
SARS-CoV-2, HRV	1 (0.7%)

**Table 3 viruses-13-00761-t003:** Results of SARS-CoV-2 detection in NPS and saliva samples collected from patients with ARI.

		NPS		Total
		Positive	Negative	
**Saliva sample**	**Positive**	51 (96.2%)	2 (3.8%)	53 (100%)
	**Negative**	5 (5.6%)	85 (94.4%)	90 (100%)
**Total**		56 (39.2%)	87 (60.8%)	143 (100%)
**Cohen’s Kappa coefficient = 0.89, 95% confidence interval 0.82–0.97,** ***p* < 0.001**				

## Data Availability

Not applicable.

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
