# Peer review of "Are Posterior Oropharyngeal Saliva Specimens an Acceptable Alternative to Nasopharyngeal Sampling for the Monitoring of SARS-CoV-2 in Primary-Care Settings?"

_viruses, 2021, doi:10.3390/v13050761_

Round 1
Reviewer 1 Report
Thank you for giving me the opportunity to review this interesting paper. I enjoyed reading it.
The authors convincingly show that saliva samples can be used for SARS-CoV-2 molecular diagnostics in primary-care setting. This is not new and several studies and metaanalyses already exist. However, this study is well designed, properly conducted and all aspects of the paper are of high quality. From this reviewer's point of view an important aspect which warrants publication is the context of ARI suveillance which could be improved by saliva samples. Moreover, the findings of other viral infections and coinfections are interesting.
There are only some minor issues I would like to ask the authors to consider in order to make the paper even more convincing:
- line 58: self-citation to reference [5] seems to be incorrect. The referenced paper is entitled "Are vaccinated measles cases protected against severe disease?". Although I cannot access the full-text in Vaccine due to a pay-wall, I assume it does not contain information about saliva for COVID-19 diagnosis. Please clarify.
- line 142-143: "34 years" -> table 1: "35"; possibly a rounding error; please correct. The same issue with "n=31" on line 143 -> table says "30".
- Figure 1 contains some typographical errors: "with drawal" -> withdrawal; "Exclusion of with antigenic results" -> what is the meaning of this statement. Why were these patients excluded at this stage (due to an antigen test result)? This needs clarification. Moreover, I would suggest to try to improve the legibility of the font in this figure - especially in the boxes with blue background.
- Table 1, expression "saliva swabbing" seems unusual since no swabs are involved in this sampling method -> would "saliva sampling" be more appropriate?
- Table 3, left heading, the expression "Sa- liva sam- " is only partially readable in my PDF-version.
- Figure 2, line 179: the authors should consider citing the R-function (lm()?) which was used to calculate the linear regression model and confidence intervals here, instead of "abline". Please also mention in the caption what the blue-shaded regions mean around the regression lines.
- Figure 2, typo: correct to "At least one respiratory virus"
Reviewer 2 Report
In this paper the usefulness of posterior oropharyngeal saliva samples for RT-PCR-based diagnosis of SARS-Cov-2 infection in primary healthcare centers was evaluated in samples tested positive for this virus by RT-PCR in NPS samples. The authors also evaluated the usefulness of posterior oropharygeal saliva for detection of other endemic and pandemic respiratory viruses. Although the number of SARS-CoV-2 infected positive patients is small the work is well done and is relevant for the field.
I have some minor comments
Figure 1- Why the exclusion of 7 patients (the word patient is missing in the last box of the figure) with antigenic tests?
Figure 2- Please indicate the rational for showing the Ct values for HRV and not any other respiratory virus? In the legend of the figure please indicate what are the lines and the intervals shown in the figures. Regarding SARS- cov2, based on the literature (See for instance Sethuraman et al, JAMA, 2020; doi:10.1001/jama.2020.8259) one would expect to find a decrease in the viral load with time and that is not what we see in this figure. It would be important to know whether this also happened for NPS samples or if this observation is specific for saliva samples.
Lines 218-19- The authors state that saliva coughed up from the posterior oropharynx seems to be much more sensitive than saliva from the oral cavity alone or from salivary glands. Regarding this issue I would suggest reading recent meta-analysis by Moreira et al. 2021 (PMID: 33670020) and by Khiabani et al, 2021 (PMID: 33774101).
